# Stressor-Specific Sex Differences in Amygdala–Frontal Cortex Networks

**DOI:** 10.3390/jcm12030865

**Published:** 2023-01-22

**Authors:** Zoé Bürger, Veronika I. Müller, Felix Hoffstaedter, Ute Habel, Ruben C. Gur, Christian Windischberger, Ewald Moser, Birgit Derntl, Lydia Kogler

**Affiliations:** 1Department of Psychiatry and Psychotherapy, Tübingen Centre for Mental Health (TüCMH), Medical Faculty, University of Tübingen, 72076 Tübingen, Germany; 2Institute of Neuroscience and Medicine, INM-7, Research Centre Jülich, 52425 Jülich, Germany; 3Institute of Systems Neuroscience, Medical Faculty, Heinrich Heine University, 40225 Düsseldorf, Germany; 4Department of Psychiatry, Psychotherapy and Psychosomatics, RWTH Aachen University, 52074 Aachen, Germany; 5JARA BRAIN Institute I, Translational Brain Medicine, 52428 Jülich, Germany; 6Neuropsychiatry Division, Department of Psychiatry, University of Pennsylvania, Philadelphia, PA 19104, USA; 7High-Field MR Center, Medical University of Vienna, 1090 Vienna, Austria; 8Centre for Medical Physics and Biomedical Engineering, Medical University of Vienna, 1090 Vienna, Austria; 9LEAD Graduate School and Research Network, University of Tübingen, 72074 Tübingen, Germany

**Keywords:** resting-state functional connectivity, sex differences, stress, social exclusion, achievement stress, Cyberball, MIST

## Abstract

Females and males differ in stress reactivity, coping, and the prevalence rates of stress-related disorders. According to a neurocognitive framework of stress coping, the functional connectivity between the amygdala and frontal regions (including the dorsolateral prefrontal cortex (dlPFC), ventral anterior cingulate cortex (vACC), and medial prefrontal cortex (mPFC)) plays a key role in how people deal with stress. In the current study, we investigated the effects of sex and stressor type in a within-subject counterbalanced design on the resting-state functional connectivity (rsFC) of the amygdala and these frontal regions in 77 healthy participants (40 females). Both stressor types led to changes in subjective ratings, with decreasing positive affect and increasing negative affect and anger. Females showed higher amygdala–vACC and amygdala–mPFC rsFC for social exclusion than for achievement stress, and compared to males. Whereas a higher amygdala–vACC rsFC indicates the activation of emotion processing and coping, a higher amygdala–mPFC rsFC indicates feelings of reward and social gain, highlighting the positive effects of social affiliation. Thus, for females, feeling socially affiliated might be more fundamental than for males. Our data indicate interactions of sex and stressor in amygdala–frontal coupling, which translationally contributes to a better understanding of the sex differences in prevalence rates and stress coping.

## 1. Introduction

In our daily lives, we are confronted with psychosocial and physiological stressors. Both can elicit a typical physiological stress response, that is, the activation of the hypothalamo–pituitary–adrenal (HPA) axis and thus the release of cortisol, which especially occurs in situations that are uncontrollable and threatening to the social self [1]. In an MRI environment, psychosocial stress is often induced using the Cyberball task [2] or the Montreal Imaging Stress Task (MIST, [3]). In the Cyberball task, participants are actively excluded from a ball-tossing game by peers, creating a threat to the basic social need of feeling included [4] and representing interpersonal, social stress. Contrarily, the MIST elicits achievement stress using mental arithmetics and also includes social evaluation. Although the single correlates of neural stress research have been investigated frequently (for a meta-analysis see, e.g., [5]), models on connectivity in neural stress research also exist but have gained less attention.

De Raedt and Hooley (2016) [6] proposed a neurocognitive framework for stress coping including three main regions: (1) the amygdala, (2) the dorsolateral prefrontal cortex (dlPFC), and (3) the ventromedial frontal region, including the ventral anterior cingulate cortex (vACC) and the (ventro)medial prefrontal cortex ((v)mPFC). The framework suggests that the amygdala, a region central to emotion processing [7], plays a key role in stress coping. The dlPFC, a brain region involved in attentional control, executive function, and the reappraisal of negative experiences [8], downregulates amygdala activity, thereby improving stress coping. In contrast, the activation of the ventromedial frontal region, which is involved in stress processing [9,10], attentional emotional processing [11], and reward processing [12], upregulates amygdala activity, resulting in a lower ability for stress coping. Thus, this framework specifies the effects to stress via brain connectivities.

Resting-state functional connectivity (rsFC) is of particular interest when investigating the interactions among brain areas. Therewith, rsFC can be used to determine the anticipatory effects and aftermath of stress in neural networks, including those described in the framework by de Raedt and Hooley [6]. Previous studies have reported increased rsFC of the amygdala–dlPFC and amygdala–vACC resulting from psychosocial stress (including both achievement and social stress components) 30 min after stress induction in a mixed sample of females and males [13]. In a male-only sample, an increased amygdala–vmPFC rsFC has also been reported one hour after psychosocial stress induction [10]. Additionally, cortisol responders showed a reduced amygdala–dlPFC rsFC and an increased amygdala–mPFC rsFC compared to non-responders, suggesting less effective coping with psychosocial stress in cortisol responders [13], and the administration of hydrocortisone to raise cortisol levels increased task-based amygdala–mPFC coupling in a dynamic facial expression task in a male-only sample [14]. In contrast to psychosocial stress, in the immediate aftermath of social exclusion, brain networks seem to shift towards a more vigilant state through an increase in the rsFC of the default mode network with the salience network, thereby increasing the allocation of neural resources to deal with social exclusion and potentially mobilise energy for a fight-or-flight response [4].

Sex differences are well known in terms of stress reactivity to psychosocial stress (achievement stress with a social component) and their coping mechanisms. Whereas females seem to be more susceptible to social stress [15], males seem to be more sensitive to achievement-related stress [16]. Sex differences are also apparent in the function and regulation of the HPA axis, with some studies reporting higher cortisol levels after stress induction in females, and others reporting the same in males (for a review see [17]). These sex differences in stress reactivity and coping might contribute to differences between the sexes in the prevalence of stress-related diseases [18], where females have a higher risk of suffering from affective disorders, such as depression and anxiety [19,20], whereas males are more likely to suffer from substance misuse [20,21]. To assess the sex differences in the coupling of brain regions, rsFC is of specific interest, as during the resting state, neural connectivity can be characterised beyond task-based activation, which is of particular interest when it comes to differences in abilities or preconditions between females and males. Concerning the sex differences in the rsFC of the amygdala with frontal regions independent of stress reactivity, several studies reported significant findings. In adolescent boys, increased left amygdala–mPFC rsFC was reported, whereas girls showed increased right amygdala–mPFC rsFC [22]. In adults, higher amygdala–vmPFC rsFC emerged in males compared to females [23], and negative associations of cortisol with the rsFC of the amygdala and the dlPFC and the subgenual ACC (sgACC), a subpart of the vACC, in females but positive associations in males were reported [24]. So far, to the best of our knowledge, no studies have specifically examined the sex differences in rsFC in situations of social exclusion or achievement stress.

Taken together, the previous studies indicate that amygdala–frontal neural networks seem to be modified by diverse stressors in specific manners and additionally are wired differently in females and males. With the current study, we set out to investigate the effect of sex and stressor type on the networks between the amygdala and frontal rsFC, following the assumption that the amygdala has a crucial part in stress processing and interacts with the frontal cortex in terms of stress coping [6]. Thus, we compare amygdala–frontal rsFC before and after two different psychosocial stressors and additionally investigate sex differences. We hypothesise that both stressors affect amygdala–frontal rsFC, but that these effects are different for females and males. More specifically, we expect higher amygdala–mPFC rsFC in males than females, in general [23], and an increase in rsFC pre- to post-stress, at least in males [10,14], whereas we cannot make any predictions for females, as this has not been previously assessed in female participants. Due to task-related sex differences, with females being more affected by social exclusion [15] and males by achievement stress [16], we additionally expect sex differences in rsFCs depending on the stressor type. Additionally, we hypothesise that stress-induced cortisol changes are associated with an increase in amygdala–mPFC rsFC in both sexes [10,13] and a decrease in amygdala–dlPFC and amygdala–vACC rsFC, especially in females [24].

## 2. Materials and Methods

### 2.1. Sample

For this study, 77 healthy, non-smoking, right-handed university students (40 females) were included. The sample size was calculated using G*Power [25] (Heinrich Heine Universität, Düsseldorf, Germany) (alpha error rate 0.05, power 0.95, f = 0.18), which calculates a total sample size of n = 68 for a repeated measurements, within-between-subjects design. Considering dropouts, we aimed to include 80 participants, 40 females and 40 males, in the study. Recruitment was carried out through advertisements posted at the Medical University of Vienna and the University of Vienna, Austria, as well as through various online student platforms. Psychology students were excluded due to their potential familiarity with the stress-induction tasks. Exclusion criteria were a history of neurological or mental disorders (assessed via a structured clinical interview, SCID, [26]), chronic illnesses, drug or hormone intake including hormonal contraception, alcohol abuse or addiction, working night shifts, engaging in competitive sports, recent or current pregnancy, premenstrual dysphoric disorder, allergic asthma, and MRI incompatibility (i.e., metal parts in the body, etc.). Participants were asked to refrain from alcohol consumption and exercise 24 h prior to testing and medication, caffeine, and drug intake on the day of testing, as well as food or drink intake (except water) two hours before arriving at the lab. We included female participants over the whole menstrual cycle to include a variety of female hormonal profiles. The cycle phase was assessed through documentation of the previous three cycle phases and confirmed via baseline measurements of estradiol and progesterone with saliva samples on the day of testing. Females and males were matched for age.

Written informed consent was obtained from all participants and the study was approved by the Institutional Review Board of the Medical University of Vienna. All participants were treated according to the Declaration of Helsinki (1964). At the end of the testing day, participants were fully briefed on the aims of the study and received financial compensation.

### 2.2. Procedure

All measurements were performed at the high-field MR Centre at the Medical University of Vienna, Austria, between 1:30 p.m. and 6:30 p.m. to control for circadian hormone rhythms. After arrival, participants received detailed instructions, filled in questionnaires pertaining to social demographics (i.e., sex, age, education status), verbal intelligence (Mehrfachwortschatztest-B (MWTB) [27]), social support in the preceding 10 days, rejection sensitivity (Rejection Sensitivity Questionnaire (RSQ) [28]), stress coping (Stressverarbeitungs-Fragebogen (SVF) [29]), and subjective ratings (positive and negative affect scale (PANAS) [30]; emotional self-rating (ESR) [31]). Figure 1 illustrates the procedure of the fMRI and saliva sample data collection.

First, the participants provided an initial saliva sample for hormone analysis (T0; approximately 15 min after arrival and before scanner entry) to analyse estradiol, progesterone, testosterone, and cortisol. The participants then underwent two fMRI sessions, with two separate tasks (achievement stress and social exclusion, described below in more detail and see also [32,33]) applied in a randomised order across the participants. Before and after each task, a resting state (rs) scan was conducted (i.e., four rs scans in total; Figure 1). Anatomical images were assessed at the beginning of the first fMRI session. To examine the effects of the tasks on the subjective ratings and hormonal stress reactivity, before the first and third rs scans and after each task, participants again completed the PANAS and ESR and provided saliva samples (Figure 1, T1–T4). Between both tasks, a break of at least 60 min was spent outside the scanner.

### 2.3. Stress Tasks

#### 2.3.1. Modified Montreal Imaging Stress Task—MIST

For achievement stress, we used a modified version of the MIST [3] in which the participants had to solve mental arithmetic problems. The task’s difficulty was adapted to each participant’s performance and the participants received feedback on their accuracy in the arithmetic tasks. The modified MIST consists of two conditions: (1) in the control condition, participants performed arithmetic tasks without time pressure, and (2) in the stress condition, participants’ success rates were kept at 20–45% by increasing the difficulty and decreasing the available time. For the current study, the MIST was modified to exclude the social component of this task by giving no negative social feedback or feedback about the performance of an average group of peers. This was necessary to compare social exclusion and achievement stress between females and males in the current study design. In total, the task lasted 10 min, with two control conditions and two stress conditions of 70 s each, repeated twice. For a more detailed description of this task, see [32].

#### 2.3.2. Cyberball

For social exclusion, we used a modified version of the Cyberball task [2], a virtual ball-tossing game played with two other players (1 male, 1 female; modified version according to [34]). The participants were instructed that they would engage in a ball-tossing game with other participants and that they were not to meet the other players before the game to avoid first impressions influencing gameplay. To increase the belief that the other players were “real”, the participants were told that they would be allowed to meet the other players after the game. Thus, the participants did not know that the two other players were computer determined. The participants had the choice of pressing one of two buttons to toss the ball to either of the other two players. The players were represented by silhouettes from a pre-recorded video of real people. The participant was represented with two hands at the bottom of the screen. The task consisted of three conditions: (1) in the technical exclusion condition, a message appeared that the network connection was not working properly; (2) in the social exclusion condition, the participants did not receive any passes, with a message assuring them that the network connection was working properly; and (3) in the inclusion condition, the participants received a third of the passes. The task started with a technical exclusion block, followed by three inclusion blocks and five exclusion blocks, and finished with two more inclusion blocks (30–40 s/block). Each block consisted of approximately 12 passes. Inter-block intervals with a fixation cross lasted 1–3 s. The total duration of the task was approximately 10 min. For a more detailed description of this task, see [33].

### 2.4. Saliva Samples

To assess hormone concentrations, saliva samples (SaliCap collection devices, Immuno-Biological Laboratories GmbH, Hamburg, Germany) were obtained at five different time points (T0–T4) four times (pre- vs. post-stressor samples) while participants were lying in the MRI scanner (see Figure 1 and Section 2.2). The samples were stored at −20 °C until shipping to the analysis laboratory (SwissHealthMed, Aying, Germany), where they were frozen again overnight at −20 °C, thawed, and centrifuged. Hormone concentrations of cortisol, testosterone, and progesterone were assessed at all time points using competitive luminescence immunoassay (LUMI) kits, whereas estradiol levels were assessed with enzyme-linked immunoassay (ELISA) kits (SwissHealthMed, Aying, Germany), as they show better sensitivity than the LUMI kits for estradiol. Estradiol was only assessed in females on arrival (T0), as it merely served to determine the menstrual cycle phase. Reliable and valid measurements were achieved for the four hormones with these kits (cortisol: intra-assay coefficient of variability (CV) < 4% and inter-assay CV < 5%; testosterone: intra-assay CV < 4% and inter-assay CV < 7%; progesterone: inter-assay CV < 4% and inter-assay CV < 5%; estradiol: intra-assay CV < 8% and inter-assay CV < 4%).

### 2.5. Data and Statistical Analysis of Behavioural and Hormonal Data

Statistical analysis of the sex differences in the demographic, subjective, and hormonal data was performed with IBM SPSS Statistics for Windows version 27 (IBM Corp., Armonk, NY, USA). For the demographic data used for the sample description (see Section 3.1), we tested the assumptions for parametric testing, and *t*-tests or non-parametric equivalents were performed. For subjective (positive and negative affect, anger) and hormonal (cortisol) stress reactivity, four separate three-factor ANCOVAs with two within-subject factors (task (achievement stress/social exclusion), time (pre-/post-stress)), one between-subjects factor (sex (female/male)), and a covariate of no interest (order of task presentation) to account for a potential effect of the order of the task presentation, were performed. The significance level for all the statistical tests was set at *p* < 0.05. ANCOVAs were Bonferroni corrected.

### 2.6. rsFC Analyses

#### 2.6.1. Definition of Regions of Interest

To assess the amygdala–frontal rsFC, we used the left and right amygdala [35] and assessed their connectivity with the following frontal areas as the regions of interest (ROIs). Based on previous literature, the unilateral vACC [36], the unilateral mPFC [37], and the left and right dlPFC [36] were chosen. As performed previously, the ROIs were created by adding a 6 mm sphere around the MNI coordinates [38] (see Figure 2 and Table 1 for ROIs and coordinates).

#### 2.6.2. Acquisition, Pre-Processing, and Calculation of rsFC

Functional and anatomical data were acquired on a 3 T TIM Trio Scanner (Siemens Healthineers, Erlangen, Germany) at the Medical University, Vienna. The anatomical images were acquired using an MPRAGE sequence (3D Magnetisation Prepared Rapid Gradient Echo: 1 × 1 × 1.1 mm resolution, TR = 2300 ms, TE = 4.21 ms, flip angle 9°, inversion time 900 ms, 160 sagittal slices). For the rs scans, a gradient-echo EPI sequence (with distortion correction; 23 interleaved slices, TE/TR = 38/1800 ms, voxel size 1.5 × 1.5 × 3 mm, 90° flip angle; bandwidth = 1446 Hz/pixel, 1.8 mm slice gap) was applied, acquiring 167 images in an axial plane for each subject. These settings produce a high spatial resolution, which, in combination with low slice thickness, helps to avoid signal dephasing in the ventral brain [39,40]. Thus, the applied parameters led to high sensitivity and specificity, especially around the amygdala.

Functional data were processed using SPM12 (Wellcome Trust Centre for Neuroimaging, London, UK, http://www.fil.ion.ucl.ac.uk/spm/software/spm12/, accessed on 1 December 2022) implemented in Matlab (Version R2019a; Mathworks Inc., Sherborn, MA, USA). The first four volumes were discarded for each participant before further processing. Then, the images were realigned to the initial image, followed by alignment to the mean EPI. The resulting mean EPI image of each subject was then spatially normalised to the ICBM-MNI152 template for each participant [41] using a “unified segmentation” approach [42]. The deformation resulting from this method was applied to the individual EPI volumes, which were further resampled to 2 mm isotropic. Smoothing was performed by a 5 mm full-width-at-half-maximum Gaussian kernel to enhance the signal-to-noise-ratio, as well as to compensate for anatomical variations. The time course of each ROI was extracted for each participant by computing the first eigenvariate of all voxels within a 6 mm sphere around the respective ROI coordinate, as performed previously [38]. These time courses were further denoised (to reduce spurious correlations [43]) by excluding the variance explained by the following variables: (a) the six motion parameters derived from the image realignment and (b) their first derivatives, as well as (c) the white matter (WM) and cerebral blood flow (CBF) intensity (each tissue signal class related to a separate signal). All these nuisance variables entered the model as first- and second-order terms to increase the sensitivity and specificity of the analyses, as well as to detect valid correlations and anti-correlations during rest [44]. Data were band-pass filtered with cutoff frequencies of 0.01 and 0.08 Hz and the rsFC of each participant was adjusted for the effects of age via a regression analysis to account for even subtle changes in brain architecture (see also [5,24]). The linear (Pearson) correlation coefficients of the resulting time series between the time series of the left and right amygdala and the time series of the four frontal ROIs (unilateral vACC and mPFC, bilateral dlPFC, see Table 1) were calculated to quantify the rsFCs, resulting in eight rsFC assessments. This was performed for each of the four rs scans (T1–T4). The resulting correlation coefficients were Fisher z-transformed for building an approximately normal distributed variable for further statistical analyses.

#### 2.6.3. Statistical Analyses of rsFCs

For this data analysis, all further statistical analyses were performed with IBM SPSS 27 (IBM Corp., Armonk, NY, USA). The rsFCs of both amygdalae with frontal ROIs were analysed using a seed-to-seed analysis method, excluding the rsFCs between the frontal ROIs or between the amygdalae, as the framework of de Raedt and Hooley (2016) [6] does not include interactions between these rsFCs. First, all eight included rsFCs (bilateral amygdala with unilateral vACC and mPFC and bilateral dlPFC) for each of the four rs scans (T1–T4) were tested for significant deviations from zero by one-sample *t*-tests. Only the rsFC values that were significantly different from zero in at least one of the four rs scans (T1–T4) were included for further analyses. This procedure excluded the rsFC of the right amygdala with the dlPFC bilaterally (all ps > 0.077), as well as the rsFC between the left amygdala and the right dlPFC (ps > 0.068). For each of the remaining five rsFCs (bilateral amygdala with unilateral vACC and mPFC; left amygdala with left dlPFC), separate three-factor ANCOVAs with two within-subject factors (task (achievement stress/social exclusion), time (pre/post-stress)), one between-subjects factor (sex (female/male)), and a covariate of no interest (order of task presentation) to account for a potential effect of the order of the task presentation were performed. Sticking to de Raedt and Hooley’s framework [6], no dependencies between rsFCs are assumed, thus for our analyses using this specific framework, we assumed the independence of the connectivities. Nevertheless, to account for repeated testing due to laterality, Bonferroni correction was applied for bilateral amygdala analyses (bilateral amygdala with unilateral vACC and unilateral mPFC), with an α-value set to 0.025. All ANCOVAs were Bonferroni corrected. Partial eta-squared as the effect size, reflecting the proportion of variance due to a certain parameter or set of parameters in a model relative to the variance in a simpler, nested model [45], is reported for ANCOVAs.

##### Exploratory Regression Analyses

To further exploratorily assess the associations of stress-induced changes in cortisol reactivity and subjective ratings with changes in the amygdala–frontal rsFC before vs. after each task, we performed multiple regression analyses. Therefore, the changes from pre-stress to post-stress in the amygdala–frontal rsFC, cortisol, positive affect, negative affect, and anger for each stressor (achievement stress/social exclusion) were calculated. For each amygdala–frontal rsFC, exploratory multiple regression analyses (with forced entry) were performed. For the following predictors, we ran separate regression analyses, with the change in the amygdala–frontal rsFC from pre-stress to post-stress as the dependent variable: change from pre- to post-stress in (a) cortisol, (b) positive affect, (c) negative affect, and (d) anger. These analyses were done separately for social exclusion and achievement stress. For each analysis, sex and order of task presentation were included as additional predictors. One exception was the change in cortisol, where we observed a significant sex difference (see Section 3.2 and Figure 3A). We, therefore, added the interaction of sex and cortisol as a predictor to the regression and found it to be significant (*p* = 0.021) and thus performed this regression analysis separately for females and males. For all regression analyses, outliers were identified and excluded using a Cook’s distance above 0.5 as the cutoff. Significance levels for all regression analyses were Bonferroni corrected for laterality, as explained above.

## 3. Results

### 3.1. Sample Description

The groups were matched for movement parameters (FD, DVARS, RMS, [44,46]), removing four male participants from the analysis and leading to a final sample of 73 participants (40 females). After matching, females and males did not differ in their mean movement (all ps > 0.56). Females and males did not differ significantly in age, verbal intelligence, amount of social support in the preceding 10 days, rejection sensitivity, or positive coping strategies (see Table 2). Besides the expected sex differences in the progesterone and testosterone values, with higher progesterone in females and higher testosterone in males, there was a significant difference in negative coping strategies, with females scoring higher (i.e., they used more negative coping strategies) than males. Almost half of the females were tested during menstruation in the early follicular phase (day 1–5, n = 19), two were tested during ovulation (day 12–14, n = 2), and the rest were tested during the midluteal phase (day 18–23, n = 19). With regard to the order of the task presentation, 19 females and 15 males first performed the achievement stress task vs. 21 females and 18 males, who first underwent social exclusion (χ^2^(1) = 0.03, *p* = 0.862, φ = 0.02). Table 2 shows the sample description.

### 3.2. Associations between Stress and Cortisol

The three-factor ANCOVA (within-subject factors: task (achievement stress/social exclusion), time (pre/post-stress); between-subjects factor: sex (female/male); covariate of no interest: order of task presentation) showed a main effect of sex (F(1,70) = 9.085, *p* = 0.004, ηp^2^ = 0.115), with higher cortisol levels in males than in females (Figure 3A). A task-by-time interaction was found (F(1,70) = 5.301, *p* = 0.024, ηp^2^ = 0.070) and post hoc *t*-tests showed significantly lower cortisol levels post- compared to pre-social exclusion (t(71) = 2.579, *p* = 0.012), which was not seen post- compared to pre-achievement stress (Figure 3B). Directly comparing the pre- and post-values between the two tasks, however, did not yield significant differences (all ps > 0.077). Figure 3A,B and Table A1 show the detailed cortisol levels.

### 3.3. Associations between Stress and Subjective Rating

For each of the three subjective rating measures, positive affect, negative affect, and anger, a three-factor ANCOVA (within-subject factors: task (achievement stress/social exclusion), time (pre-/post-stress); between-subjects factor: sex (female/male); covariate of no interest: order of task presentation) was performed.

*Positive affect.* We observed a significant main effect of time (F(1,71) = 21.035, *p* < 0.001, ηp^2^ = 0.229), with lower positive affect ratings post-stress compared to pre-stress. No other main or interaction effects reached significance (all ps > 0.386, see Figure 3C and Table A1).

*Negative affect.* A main effect of task (F(1,71) = 5.896, *p* = 0.018, ηp^2^ = 0.077), with higher negative affect for achievement stress than for social exclusion, and a main effect of time (F(1,71) = 10.358, *p* = 0.002, ηp^2^ = 0.127), with higher negative affect post-stress compared to pre-stress, were found. No other main or interaction effects were seen (all ps > 0.071, see Figure 3D and Table A1).

*Anger.* Anger showed a main effect of task (F(1,71) = 11.849, *p* < 0.001, ηp^2^ = 0.143), with higher ratings for achievement stress compared to social exclusion, and a main effect of time (F(1,71) = 45.125, *p* < 0.001, ηp^2^ = 0.389), with higher ratings post-stress compared to pre-stress. Further, a task-by-time interaction could be seen (F(1,71) = 14.689, *p* < 0.001, ηp^2^ = 0.171). Post hoc *t*-tests showed that anger was significantly higher after both social exclusion (t(72) = −2.819, *p* = 0.006) and achievement stress (t(72) = −6.158, *p* < 0.001), but was higher after achievement stress than after social exclusion (t(72) = −4.125, *p* < 0.001). No significant effects were seen before the tasks, nor were there effects of sex or other interaction effects (all ps > 0.085, see Figure 3E and Table A1).

### 3.4. Associations between Stress and rsFC

For each of the five retained rsFCs, a three-factor ANCOVA (within-subject factors: task (achievement stress/social exclusion), time (pre/post-stress); between-subjects factor: sex (female/male); covariate of no interest: order of task presentation) was conducted.

#### 3.4.1. Amygdala–vACC

*Right amygdala–vACC*. The ANCOVA showed a significant task-by-sex interaction (F(1,69) = 5.533, *p* = 0.022, ηp^2^ = 0.074). Post hoc *t*-tests demonstrated higher rsFC for social exclusion than for achievement stress in females (t(39) = 2.210, *p* = 0.033), whereas this difference was not apparent in males (*p* = 0.200). Furthermore, females exhibited higher rsFC than males for social exclusion (t(70) = −2.068, *p* = 0.042, see Figure 4A), which was not apparent for achievement stress. No other main or interaction effects were significant (all ps > 0.126).

*Left amygdala–vACC*. No significant effects emerged considering the Bonferroni correction (all ps > 0.039, cutoff α = 0.025).

#### 3.4.2. Amygdala–mPFC

*Right amygdala–mPFC.* A task-by-sex interaction (F(1,69) = 6.093, *p* = 0.016, ηp^2^ = 0.081) emerged. Post hoc *t*-tests indicated higher rsFC for social exclusion in females than males (t(71) = −2.543, *p* = 0.013, see Figure 4B). No significant sex difference appeared for achievement stress and no task differences appeared within sexes (all ps > 0.051). No other main or interaction effects were significant (all ps > 0.079).

*Left amygdala–mPFC*. Here, a main effect of time was observed (F(1,69) = 6.542, *p* = 0.013, ηp^2^ = 0.087), with higher rsFC pre-stress compared to post-stress. No other main or interaction effects were significant (all ps > 0.069).

#### 3.4.3. Amygdala–dlPFC

For the rsFC of the left amygdala with the left dlPFC, no significant effects emerged (all ps > 0.081).

### 3.5. Cortisol and Subjective Ratings as Predictors of rsFC Changes

To assess the impact of subjective and physiological stress reactivity on rsFC, we conducted exploratory multiple regression analyses. Therefore, the changes in the subjective ratings and cortisol levels from pre- to post-stress were calculated separately for each paradigm and used as predictors of the changes in the rsFCs from pre- to post-stress. Sex and order of task presentation were additionally added as predictors. Table 3 shows the values of the changes in cortisol and subjective ratings that were used for analysis.

#### 3.5.1. Predictor: Changes in Cortisol from Pre- to Post-Stress

For the changes in cortisol from pre- to post-stress, we ran separate analyses for females and males, as we observed differences in the cortisol levels between the sexes in the current sample, and the interaction of sex with cortisol turned out to be significant in the regression (predictors: change in cortisol from pre- to post-stress and order of task presentation).

*Females, right amygdala–vACC; social exclusion.* For social exclusion, the cortisol changes were significantly positively associated with the changes in the rsFC (R^2^ = 0.204, F(2,36) = 4.624, *p* = 0.016). A decrease in cortisol during social exclusion was associated with a decrease in the rsFC between the right amygdala and vACC (β = 0.377, t = 2.513, *p* = 0.017; see Figure 4C). No other significant associations emerged (all ps > 0.066).

The changes in cortisol from pre- to post-stress did not predict changes in any other rsFCs either for social exclusion or achievement stress for both sexes (all ps > 0.102).

#### 3.5.2. Predictor: Changes in Positive Affect from Pre- to Post-Stress

For the changes in the positive affect from pre- to post-stress, multiple regression analyses were performed with the following predictors: change in positive affect, sex, and order of task presentation.

*Left amygdala–mPFC, achievement stress.* The changes in the positive affect had a significant negative relationship with the changes in the rsFC (model R^2^ = 0.132; F(3,69) = 3.497, *p* = 0.020) for achievement stress across both sexes. A decrease in the positive affect due to achievement stress was associated with an increase in the rsFC (β = −0.227, t = −2.003, *p* = 0.049, see Figure 5A).

The changes in the positive affect from pre- to post-stress did not predict changes in any other rsFC either for social exclusion or achievement stress (all ps > 0.051). 

#### 3.5.3. Predictor: Changes in Negative Affect from Pre- to Post-Stress

For the changes in the negative affect from pre- to post-stress, regression analyses were performed with the following predictors: change in negative affect, sex, and order of task presentation. The changes in the negative affect from pre- to post-stress did not predict changes in any other rsFC either for social exclusion or achievement stress (all ps > 0.045, cutoff α = 0.025).

#### 3.5.4. Predictor: Changes in Anger from Pre- to Post-Stress

For the changes in anger from pre- to post-stress, regression analyses were performed with the following predictors: change in anger, sex, and order of task presentation.

*Left Amygdala–vACC, achievement stress*. The changes in anger were negatively associated with the changes in the rsFC (model R^2^ = 0.139; F(3,69) = 3.699, *p* = 0.016); with increased anger, decreased rsFC was seen for achievement stress across both sexes (β = −0.336, t = −3.005, *p* = 0.004), see Figure 5B.

The changes in anger from pre- to post-stress did not predict changes in any other rsFC either for social exclusion or achievement stress (all ps > 0.083).

## 4. Discussion

The aim of the current study was to investigate the effects of sex and stressor type on the rsFC of the amygdala with frontal regions. This selection of regions was based on a recent neurocognitive framework on stress coping by de Raedt and Hooley [6]. We discuss our results below.

### 4.1. Sex Differences in Stress Reaction in Amygdala–Frontal Stress Networks

We observed significant sex differences in the amygdala–frontal network, with higher amygdala–vACC and amygdala–mPFC rsFCs for social exclusion in females, which was not seen in males. Additionally, in females, the rsFC of the amygdala and the vACC was significantly higher for social exclusion than for achievement stress. Furthermore, in an exploratory multiple regression analysis, females showed an association between changes in cortisol values from pre- to post-stress with the rsFC between the amygdala and the vACC. Hence, social exclusion seems to exhibit a sex-specific effect on the amygdala–frontal network. This fits nicely with previous reports, which show that females are more susceptible to social affiliation and exclusion [15,16]. For the current results, besides directly experiencing social exclusion, females might also be more affected than males by the anticipatory effects of an upcoming social, interpersonal stress situation, as participants were told that they would be participating in a ball-tossing game. In previous studies, increased amygdala–vACC connectivity was found in trauma-exposed females at rest and during affective processing [47], during implicit processing of sad facial stimuli in healthy females and males [48], and following a conflict task using emotional stimuli [49]. This supports the assumption in the neurocognitive framework [6] that the activation of the ventromedial frontal region, which is involved in stress [9,10] and attentional emotional processing [11], upregulates amygdala activity, further resulting in a lower ability for stress coping. The vACC is further closely related to stress processing after social exclusion [9] and social decision making. Importantly, the vACC coordinates used in the current study (based on [36]) are more specifically located in a part of the vACC known as the subgenual ACC (sgACC) [50]. In previous studies, the sgACC was often associated with mood disorders such as depression, showing a reduced grey matter volume and higher activation of the sgACC in depressed compared to healthy participants [51], as well as with the processing of sadness [52,53]. A heightened amygdala–vACC rsFC in social exclusion compared to achievement stress in females is in line with the aforementioned higher susceptibility of females to social exclusion and suggests more affective processing in socially stressful situations than in achievement-related stress in females. In our exploratory multiple regression analyses, we additionally saw a positive association between changes in cortisol and changes in amygdala–vACC rsFC in females for social exclusion, which was reported previously in a mixed sample for psychosocial stress (thus, achievement stress, including a social component) [13].

Taken together with the previous literature, the current results show that rsFC between the amygdala and the vACC is elevated in stressful situations when females are exposed to emotionally or socially stressful contexts such as social exclusion and they therewith must engage in emotion processing. As females are more susceptible to social affiliation and social exclusion, social stress might be particularly salient for them. Thus, our findings corroborate the assumption that females engage more coping mechanisms and are more affected by socially stressful situations than males. In addition, they are more triggered in socially stressful situations than in achievement-stress situations.

Moreover, in the current results, we saw a higher coupling between the amygdala and mPFC for social exclusion for females compared to males. Increased amygdala–mPFC rsFC was previously observed in a male-only sample after psychosocial stress (achievement stress, also including a social component) [10], in reward learning during wins compared to losses in both sexes [12], and has been associated with lower anxiety [54]. Thus, an elevated rsFC between the amygdala and the mPFC is present in situations of positive gain and achievement-related stress situations, at least in males. Together with the current results, this suggests that females possibly experience social gain from interactions, although in stressful situations. Indeed, reward anticipation, as in this case, the social interaction for females, could even buffer subjective and hormonal stress reactivity [55].

In addition to this, the observed sex difference of higher amygdala–mPFC rsFC in females compared to males is partly in contrast to a previous study, which described heightened amygdala–mPFC rsFC in males compared to females. This result is however independent of an applied task and rather describes the connectivity between the laterobasal amygdala and the rectal gyrus [23]. Furthermore, our results also emphasise the importance of setting up study designs specifically investigating sex differences and including both females and males, as male-only samples suggested heightened amygdala–mPFC rsFC in a facial expression task [14] and during the stress-recovery phase one hour after stress [10]. Taken together with the current results, this shows that neglecting females in research, biases the outcome interpretation. Thus, the investigation of sex- and stressor-specific differences and interactions is still in its infancy and more studies on sex differences in stress specifically should be performed.

In our current study, we propose that heightened rsFC of the amygdala with the vACC and mPFC before and after social exclusion in females compared to males and compared to achievement stress might indicate that females feel rewarded by social engagement and experience social gain from interactions. Additionally, females seem to be more susceptible to social stress, which might be particularly salient for them, and they are, therefore, more affected by socially stressful situations and need to engage more coping mechanisms within these situations than males.

The assessment of sex differences is particularly important with regard to the different prevalence rates of mental disorders, as women are more susceptible to stress-related disorders such as depression and anxiety [20], whereas men are more prone to suffer from substance misuse [20,21]. These prevalence rates might be affected by different stressor types, such as social exclusion or achievement stressors, in sex-dependent, distinct ways. Sex-specific reactions to different stressors might partly explain the sex-specific prevalence rates in mental disorders [18].

### 4.2. Sex-Independent Effects of Stress Reaction on Amygdala–Frontal Stress Networks

The stressor type seems to be influential not only regarding sex differences but also regarding the general differences that appeared in both females and males for social exclusion and achievement stress. Cortisol levels decreased after stress induction in social exclusion but not following stress induction in achievement stress. Anger showed a stressor-specific effect with higher anger ratings post-achievement stress that was not present post-social exclusion. Further, bearing in mind that our multiple regression analyses were merely exploratory, we found negative associations of positive affect and anger with amygdala–frontal rsFCs in achievement stress: decreased positive affect was associated with an increased amygdala–mPFC rsFC and decreased anger was associated with an increased amygdala–vACC rsFC.

Although we were expecting a cortisol increase after the stressors and did not see it, this lack of cortisol increase has been reported repeatedly in other studies after stress induction in an fMRI environment ([56], for a review see [57]). The fMRI environment itself can be stressful and lead to high cortisol levels preceding the scan, which then normalise after the scan and in subsequent scans [58]. Generally, within the circadian cortisol rhythm, levels decrease during the course of the day, referred to as the diurnal cortisol slope [59]. This diurnal decrease was not apparent for achievement stress in the current study, indicating some (weak) cortisol reactivity. Interestingly, we also did not see any changes in amygdala–dlPFC rsFC for social exclusion or achievement stress, which might be explained by the missing cortisol increase in our participants. Based on the neurocognitive framework by de Raedt and Hooley, increased connectivity between the amygdala and the dlPFC is associated with successful stress coping. Nevertheless, Quaedflieg and colleagues [13] reported reduced amygdala–dlPFC coupling only in cortisol responders compared to non-responders, suggesting worse stress coping in cortisol responders than non-responders. Thus, the coupling of the amygdala and dlPFC might be connected to hormonal changes such as cortisol expression [24]. Nevertheless, even with the missing cortisol increase, the subjective parameters, such as elevated anger and negative affect, as well as decreased positive affect, clearly indicate an increase in stress and negative emotions. Anger, in particular, showed a stressor-specific increase, with higher ratings post-achievement stress compared to post-social exclusion. Together with the exploratory results of a decreasing amygdala–vACC rsFC with increasing anger, this might suggest that achievement stress negatively and subjectively affected the participants. Higher rsFC of the amygdala with the vACC was previously found in situations of achievement stress with an additional social component [10,13], but not in achievement stress alone. Further, we found that decreased positive affect was associated with increased amygdala–mPFC rsFC. Taken together with the literature showing that amygdala–mPFC rsFC was present after psychosocial stress (achievement stress with a social component) in a male-only sample [10], as well as in male and female cortisol responders compared to non-responders [13], Quaedflieg and colleagues suggested that this indicates less effective stress coping, which fits the framework of De Raedt and Hooley, as connectivity between the ventromedial frontal regions and the amygdala suggests less effective stress coping.

Taken together, achievement stress is subjectively perceived as more stressful than social exclusion, which is corroborated by increased anger, negative affect, and (according to the circadian rhythm) a missing decrease in cortisol levels. Higher anger and lower positive affect are connected to lower amygdala–vACC rsFC and higher amygdala–mPFC rsFC, showing an association between the subjective parameters and amygdala–frontal coupling.

### 4.3. Strengths, Limitations, and Future Directions

In the current study, we separately investigated two different stressors, namely social exclusion and achievement stress, in a within-subject design. We investigated rsFC before and after each of these stressors, enabling us to investigate changes in coupling between the amygdala and frontal regions due to social exclusion and achievement stress. As most studies on stress reactivity have so far investigated male-only samples or did not investigate sex-specific analyses in mixed samples, in the current study, we not only included a fair number of females and males but additionally specifically assessed sex differences in social exclusion and achievement stress, as well as in rsFC.

Previous studies showed that the menstrual cycle and hormonal contraceptive use affect stress reactivity and rsFC [60,61], which should be considered in future studies. For the current study, the sample size was too small to additionally include menstrual cycle effects in the analyses. However, as females had significantly higher values in negative stress coping strategies (see Table 2), we exploratorily performed a regression analysis to investigate whether the menstrual cycle phase predicted negative stress coping strategies. The linear regression analysis with the menstrual cycle phase as a predictor (early follicular/ovulation/midluteal phase) and negative stress coping strategies as the dependent variable yielded no significant association between negative stress coping strategies and the menstrual cycle phase (R^2^ = 0.059, F(1,38) = 2.380, *p* = 0.131). The negative stress coping strategies reported by the females in this study, therefore, do not seem to be associated with their menstrual cycle phase.

For reasons of homogeneity, the current sample comprised young university students. For generalisability of the results and regarding stress-related mental disorders, future studies should assess more heterogenous samples including different education levels throughout the lifespan.

Further, for the current study, we relied on rsFC between the amygdala and frontal regions, as suggested in the neurocognitive framework for stress coping by De Raedt and Hooley, to assess the impact of the different stressors on neural stress networks. However, as previous studies have shown that analyses of amygdala connectivity at the whole-brain or network level can also help to better understand the impact of stress and sex on rsFC [24,62], it would be of additional interest for future studies to create whole-brain functional connectivity maps for the amygdala and specifically investigate the impact of different stressors and sex differences on amygdala rsFC.

Finally, we want to acknowledge that although we investigated sex differences, we did not take gender into account. Whereas sex is usually categorised into female and male and is defined primarily based on biological attributes, gender refers to the socially constructed identity of female, male, and other gender-diverse people (SAGER guidelines, [63]). It is important to include higher gender diversity in studies to make them more broadly generalisable, as sex, as well as gender and their interaction (i.e., in transgender people), can have an influence on health and well-being in a variety of ways [63].

## 5. Conclusions

With the current study, we set out to investigate the effects of sex and stressor type on networks between the amygdala and frontal regions with rsFC, following a neurocognitive framework for stress coping with the assumption that the amygdala has a crucial part in stress processing and interacts with the frontal cortex in terms of stress coping. We observed significant sex differences for social exclusion in the amygdala–frontal network, as seen in the higher rsFC of the amygdala with the vACC and mPFC in females than in males. This relates to the higher emotions in females compared to males and affects regulation in socially stressful situations, which might be attributed to higher social gain in females compared to males.

Thus, socially stressful situations have a greater impact on neural stress networks in females than in males, and neural stress networks seem to be modified for diverse stressors in a sex-specific manner. The current results depict the neural basic principles of sex differences in different stressful situations, where females are triggered more intensely than males by social situations and may see social interactions as more rewarding but also more difficult in terms of engaging coping resources. From a translational perspective, the results represent the neural basis of the interaction between sex and stressor in stress reactions and contribute to a better understanding of this interaction, which affects sex-specific prevalence rates. This knowledge is important for understanding stress-related mental disorders, as it demonstrates the differences in the prevalence rates for females and males. Understanding and disentangling sex differences in acutely stressful situations may help to further discern long-term stress effects and their influence on mental disorders specific to females and males.

## Figures and Tables

**Figure 1 jcm-12-00865-f001:**
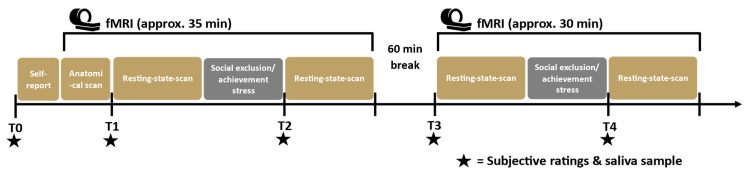
Study design. After arrival and self-report questionnaires, two fMRI sessions consisting of an anatomical scan (only before the first session), two rs scans per session, a stress induction task using either the Cyberball (for social exclusion) or modified MIST (for achievement stress) were administered to participants, with a 60 min break in between. Saliva samples and subjective ratings were collected at dedicated time points. Social exclusion and achievement stress tasks were administered in a randomised order, counterbalanced between females and males.

**Figure 2 jcm-12-00865-f002:**
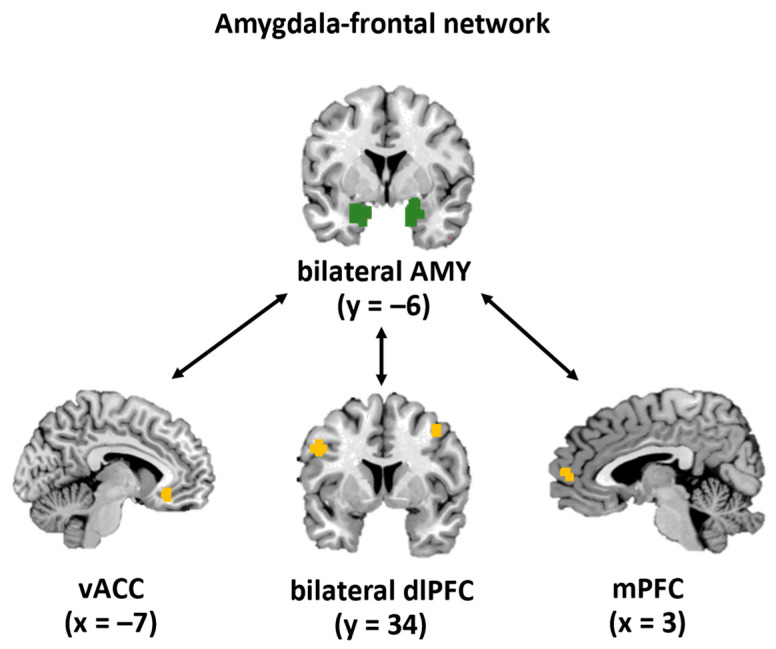
Amygdala–frontal network: representation of the ROIs. For the analyses, we assessed the rsFC of the amygdalae (in green) with one of the other ROIs (in orange), namely the ventral anterior cingulate cortex (vACC, unilaterally), the medial prefrontal cortex (mPFC, unilaterally), or the dorsolateral prefrontal cortex (dlPFC, bilaterally).

**Figure 3 jcm-12-00865-f003:**
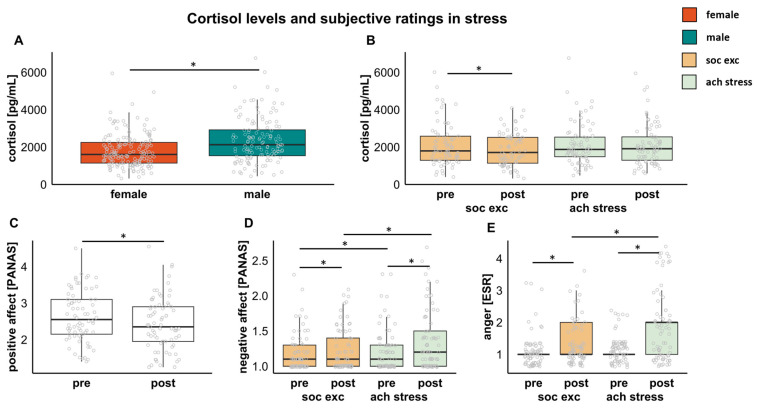
(**A**) Significantly higher cortisol level in females compared to males; (**B**) Cortisol levels for each stressor show a decrease post- vs. pre-social exclusion, which is not apparent for achievement stress; (**C**) Positive affect decreases pre- compared to post-stress in both tasks; (**D**) Negative affect is higher in achievement stress compared to social exclusion and pre- compared to post-stress; (**E**) Increased anger rates post-stress are seen in both tasks, with higher ratings post-achievement stress compared to post-social exclusion. * Indicates significance of *p* < 0.05. Abbreviations: PANAS = positive and negative affect scale, ESR = emotional scale rating, soc. exc. = social exclusion, ach. stress = achievement stress.

**Figure 4 jcm-12-00865-f004:**
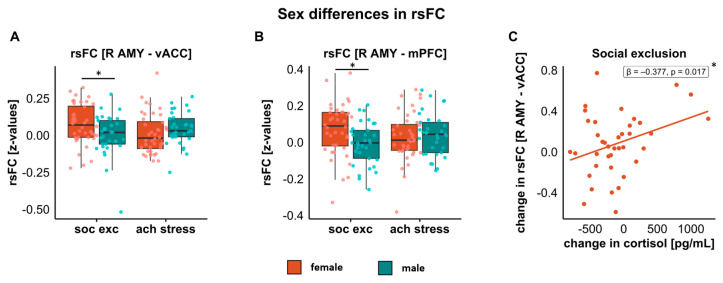
Sex differences for both paradigms in rsFC of right amygdala with vACC (**A**) and mPFC (**B**). (**C**) Association of changes in cortisol with changes in amygdala–vACC rsFC for social exclusion in females. * Indicates significance of *p* < 0.05. Abbreviations: vACC = ventral anterior cingulate cortex, AMY = amygdala, mPFC = medial prefrontal cortex, soc. exc. = social exclusion, ach. stress = achievement stress.

**Figure 5 jcm-12-00865-f005:**
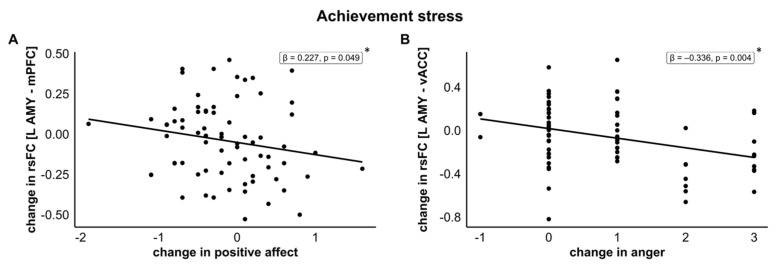
(**A**) Association between changes in rsFC of the amygdala and the mPFC and changes in the positive affect from pre- to post-achievement stress, and (**B**) association between changes in rsFC of the amygdala and the vACC and changes in anger from pre- to post-achievement stress. Change values from before to after social exclusion or achievement stress, respectively. * Indicates significance of *p* < 0.05. Abbreviations: vACC = ventral anterior cingulate cortex; AMY = amygdala; mPFC = medial prefrontal cortex.

**Table 1 jcm-12-00865-t001:** MNI coordinates of each ROI.

MNI Coordinates		X	Y	Z
Amygdala (bilateral)	R	26	−6	−14
	L	−24	−6	−14
vACC (unilateral)		−7	29	−12
mPFC (unilateral)		3	54	6
dlPFC unilateral	R	37	34	35
	L	−37	44	37

Note. Abbreviations: vACC = ventral anterior cingulate cortex, mPFC = medial prefrontal cortex, dlPFC = dorsolateral prefrontal cortex.

**Table 2 jcm-12-00865-t002:** Sample description including details of age, questionnaire data, and hormonal concentrations.

	Females (n = 40)		Males (n = 33)			
	Mean	SD	Mean	SD	Statistical Parameters	*p*-Value
Age (in years)	24.7	3.8	24.0	3.0	t(71) = 0.755	0.453
Verbal intelligence	26.7	4.0	27.4	3.5	t(70 ^1^) = 0.875	0.385
Social support in preceding 10 days	19.3	20.5	20.2	19.2	t(71) = 0.201	0.841
Rejection sensitivity	8.4	3.3	8.1	2.9	t(71) = 0.412	0.681
Positive stress coping strategies	2.2	0.4	2.3	0.3	t(71) = 0.298	0.767
Negative stress coping strategies	2.0	0.6	1.7	0.6	t(71) = 2.080	0.041 *
Estradiol levels on arrival [pg/mL]	4.58	4.07	n/a	n/a	n/a	n/a
Progesterone levels on arrival [pg/mL]	71.97	57.61	42.65	29.48	t(69 ^1^) = 2.582	0.012 *
Testosterone levels on arrival [pg/mL]	25.99	19.07	87.93	60.62	t(71) = 6.114	<0.001 *

Note. Abbreviations: SD = standard deviation. n/a = not available. * Indicates significance of *p* < 0.05. ^1^: Datapoint missing for one or two participants.

**Table 3 jcm-12-00865-t003:** Changes in cortisol and subjective ratings due to social exclusion and achievement stress.

	Social Exclusion				Achievement Stress			
	Females (n = 40)		Males (n = 33)		Females (n = 40)		Males (n = 33)	
	Mean	SD	Mean	SD	Mean	SD	Mean	SD
Change in cortisol [pg/mL]	−115.05	437.71	−373.09	1021.32	124.15	630.81	−152.91	909.76
Change in positive affect (PANAS)	−0.26	0.55	−0.24	0.57	−0.11	0.61	−0.21	0.58
Change in negative affect (PANAS)	0.03	0.25	0.05	0.25	0.19	0.51	0.11	0.30
Change in anger (ESR)	0.18	0.64	0.24	0.61	0.80	1.11	0.76	1.06

Note. Abbreviations: SD = standard deviation; PANAS = Positive and Negative Affect Scale; ESR = Emotional Scale Rating.

## Data Availability

The data presented in this study are available on request from the corresponding author. The data are not publicly available due to missing consent forms for public data sharing.

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
