# Peer review of "Stressor-Specific Sex Differences in Amygdala–Frontal Cortex Networks"

_jcm, 2023, doi:10.3390/jcm12030865_

Round 1
Reviewer 1 Report
This study investigated the neural correlates of stress during stress-inducing fMRI tasks. Specifically, they were interested in understanding potential sex differences in the patterns of brain activation that may underlie self-reported stress experiences. Several sex-specific patterns were observed that were unique to the type of stress experience. Overall, the study was well written and the data support the conclusions. Below are some minor suggestions to consider:
1) The use of the word "substance abuse" is not appropriate according to the NIDA and NIAAA. It is recommended to replace "abuse" with "misuse".
2) Given that females showed significantly higher negative stress coping strategies, and that almost half of the females were tested during the early follicular phase while the other half was tested during the mid-luteal phase, the authors should run a correlation between negative stress coping strategies and menstrual phase. Although the authors discuss the low sample sizes as a reason why they did not run additional analyses regarding the role of the menstrual cycle, a preliminary correlational analysis could provide some interesting insight.
Author Response
Please see the attachment for the response to both reviewers.

Reviewer 2 Report
The study of Bürger et al. investigated the effect of sex and stressor type in a within-subject counterbalanced design on resting-state functional connectivity (rsFC) of the amygdala with these frontal regions in 77 healthy participants (40 females). The idea of investigated the interactions of sex and stressor on resting-state functional connectivity of the amygdala with frontal regions is very good. It will certainly contribute to the existing literature and to a better understanding of the stressor-specific sex differences in amygdala-frontal cortex networks given the complex and full range statistical analysis.
My main concern is the rational of only three regions in the frontal region were selected as ROI. Even authors had explained that this is based on recent neurocognitive framework on stress coping by de Raedt and Hooley, I am still curious about who other regions that had not been present in current study. Actually one of the advantage of rsFC is the convenience of generate whole brain FC maps. And the neglect of other frontal regions is a pity. Authors can show it in the supplementary as explore analysis or at least discuss this issue in the limitation. Other minor suggestions to further improve the manuscript as below.
1. Page 5. “a covariate-of-no-interest (order of task presentation) were performed”. Why not age and education status were also used as the covariate?
2. Page 7.“Partial eta-squared as effect size is reported for ANCOVAs”. Please provide more explanations about the partial eta-squared.
3. Page 9. The authors should clarify the meaning of * in the Fig.3
4. Page 10. Figure 4 “Sex differences in rsFC of right amygdala for both paradigms with vACC (A) and mPFC (C). In (B), association of change in cortisol with change in amygdala-vACC rsFC for social exclusion in females”. The label is in the wrong position. Should add * on the scatter plot.
Author Response

(The authors gave the same response as above.)
